# Regulatory Role of microRNA of Milk Exosomes in Mastitis of Dairy Cows

**DOI:** 10.3390/ani13050821

**Published:** 2023-02-24

**Authors:** Bruno Stefanon, Michela Cintio, Sandy Sgorlon, Elisa Scarsella, Danilo Licastro, Alfonso Zecconi, Monica Colitti

**Affiliations:** 1Department of Agriculture, Food, Environmental and Animal Science, University of Udine, 33100 Udine, Italy; 2AREA Science Park, Padriciano, 99, 34149 Trieste, Italy; 3One Health Unit, Department of Biomedical, Surgical and Dental Sciences, University of Milan, Via Pascal 36, 20133 Milano, Italy

**Keywords:** milk, exosomes, miRNA, mastitis, cow

## Abstract

**Simple Summary:**

The microRNA (miRNA) cargos of exosomes isolated from milk were investigated in relation to the healthy conditions of the mammary gland. Samples were collected from a group of cows without mastitis (H), a group at risk of mastitis (ARM), and a group with subclinical mastitis (SCM). The differential expression analysis identified 38, 18, and 12 miRNAs (*p* < 0.05) in the comparisons of H vs. ARM, ARM vs. SCM, and H vs. SCM, respectively. Only bta-mir-221 was shared between the three groups. The miRNA cargos of milk exosomes can be considered to study the complex molecular machinery set in motion in response to mastitis in dairy cows and the quality of milk.

**Abstract:**

The aim of this study was to compare the cargos of miRNA in exosomes isolated from the milk of healthy (H) cows, cows at risk of mastitis (ARM), and cows with subclinical mastitis (SCM). Based on the number of somatic cells and the percentage of polymorphonuclear cells, 10 cows were assigned to group H, 11 to group ARM, and 11 to group SCM. After isolating exosomes in milk by isoelectric precipitation and ultracentrifugation, the extracted RNA was sequenced to 50 bp long single reads, and these were mapped against Btau_5.0.1. The resulting 225 miRNAs were uploaded to the miRNet suite, and target genes for *Bos taurus* were identified based on the miRTarBase and miRanda databases. The list of differentially expressed target genes resulting from the comparisons of the three groups was enriched using the Function Explorer of the Kyoto Encyclopedia of Genes and Genomes. A total of 38, 18, and 12 miRNAs were differentially expressed (DE, *p* < 0.05) in the comparisons of H vs. ARM, ARM vs. SCM, and H vs. SCM, respectively. Only 1 DE miRNA was shared among the three groups (bta-mir-221), 1 DE miRNA in the H vs. SCM comparison, 9 DE miRNAs in the ARM vs. SCM comparison, and 21 DE miRNAs in the H vs. ARM comparison. A comparison of the enriched pathways of target genes from the H, SCM, and ARM samples showed that 19 pathways were differentially expressed in the three groups, while 56 were expressed in the H vs. SCM comparison and 57 in the H vs. ARM comparison. Analyzing milk exosome miRNA cargos can be considered as a promising approach to study the complex molecular machinery set in motion in response to mastitis in dairy cows.

## 1. Introduction

The milk of cows is a source of nutrients and other compounds [1,2,3]. Moreover, breastfeeding modulates the development of the immune system [4], and this activity is also regulated by microRNAs (miRNAs) [5]. Milk is a rich source of extracellular vesicles (EVs), which are involved in cell communication [6]. Milk-derived EVs contain mRNA, miRNA, ribosomal RNA (rRNA), long noncoding RNA (lncRNA), transfer RNA (tRNA), and DNA in addition to lipids, metabolites, and proteins [7]. MiRNA in milk [8,9] is a class of single-stranded noncoding RNAs about 22 nucleotides in length that induce post-transcriptional silencing of genes by binding to the 3’ untranslated region of target genes [10]. In this way, miRNAs control multiple cellular functions, from cell differentiation to tissue development and immune regulation, and consequently many aspects of health and disease [11,12].

Mastitis is a common disease of dairy cows that causes high economic losses and often requires the administration of antimicrobial drugs [13,14], with potential implications for antimicrobial resistance [15,16,17], a topic under the scrutiny of the World Health Organization (WHO, https://www.who.int/, accessed on 14 December 2022).

Somatic cell count (SCC) in milk is the most common routine tool used at the farm level to screen cows for mastitis, and 200,000/mL SCC is the European cutoff to identify an inflammatory response to infection [18], while the cutoff is >150,000/mL SCC for New Zealand and >250,000/mL for Australia [19]. Cows with high SCC levels can be later subjected to microbial culture or molecular analysis to confirm and diagnose the type of infection and stage of disease. Recently, it has become possible to routinely calculate the percentage of polymorphonuclear neutrophils (PMN) and lymphocytes (LYM) within the SCC [20,21,22], also defined as differential somatic cell count (DSCC), which is an estimate of the percentage of polymorphonuclear cells and allows a more accurate classification of intramammary infection in dairy cows [20]. Interestingly, analysis of SCC and DSCC in milk is also related to milk quality, in particular to coagulation characteristics for cheese production [23,24].

The identification of early biomarkers of mastitis is important and forms the basis for defining and applying farm protocols to reduce the spread of infectious agents, implement hygiene measures, and thus limit the use of antimicrobial drugs. Moreover, signaling pathways and cellular functions activated in response to an inflammatory response or disease can be assessed by the study of miRNA contained in milk exosomes, which have gained attention as markers of cow health [25,26,27,28,29] and mastitis [30,31,32,33]. In fact, the availability of next-generation sequencing technology has stimulated studies that use genomics as a diagnostic tool for mastitis while enabling the molecular response of the host to microbial invasion [33]. These findings may also provide new insights for understanding the variable prevalence of mastitis among dairy cow breeds and the high individual susceptibility [34].

It has been reported that miRNAs contained in milk exosomes are conserved among mammalian species, at least among humans, pigs, cows, and pandas [35]. The miRNAs in milk exosomes are acid-resistant [36,37] and protected from degradation in the intestine [38]. Their transfer from mother to infant has been documented [39]; notably, miRNAs in milk exosomes of cow can be incorporated into human cells such as intestinal cells and macrophages [40,41,42]. Bovine exosomes isolated from milk have been used as vehicles for drug delivery for therapeutic purposes [43], confirming their uptake in the gut. These findings open the perspective of considering milk exosome loading as a newer aspect of milk safety and quality [44].

The aim of this study was to measure miRNA cargoes in exosomes isolated from milk samples of lactating cows classified as healthy or at risk or affected by mastitis based on the European cutoff of 200,000/mL SSC and the proportion of PMN. The regulatory network of differentially expressed miRNA between these groups of cows was also investigated to reveal possible signaling pathways and cellular functions activated in response to an inflammatory response or disease. Moreover, to evaluate whether the presence of an inflammatory process can influence the expression of miRNAs and milk quality, in the present study milk was collected from the all the four quarters of each cow.

## 2. Materials and Methods

### 2.1. Animals and Housing

Sixty cows were recruited for the study from a herd on a commercial farm (46.1134059, 13.2817455; N 46°6′48.261″, E 13°16′54.283″; Italy). Cows were housed in loose housing with cubicles, fed the same total mixed ration, and not treated with antibiotics during the previous 20 days. The milking parlor (parallel 12 + 12) was located next to the bedding area. All protocols, procedures, and care of the animals were in accordance with the Italian legislation on animal care (DL n.116, 27/1/1992), and the study was approved by the Ethics Committee of the University of Udine (OBPA Prot. N. 9/2020).

Before starting the sampling, milk from these cows was analyzed for SCC and DSCC values obtained from the official register of the breeders’ association (Associazione Allevatori del Friuli Venezia Giulia, Codroipo, Italy; www.aafvg.it, accessed on 14 December 2022). SCC and DSCC were measured in milk samples using a Fossomatic 7DC (FOSS Electric A/S, Hillerød, Denmark; according to ISO 13366-2/IDF 148-2:2006). The 60 cows in the herd were divided into 3 groups of 20 cows each according to the classification proposed by Zecconi et al. [24]: healthy (group H, SCC/mL < 200,000 and DSCC ≤ 69.3%); at risk (group ARM, SCC/mL < 200,000 and DSCC > 69.3%); with subclinical mastitis (group SCM, SCC/mL > 200,000 and DSCC > 69.3%). Milk samples were collected again during the subsequent monthly official record of the breeders’ association. Based on the analysis of SCC and DSCC of the second sampling, the 60 cows were reclassified as follows: 34 cows were in group H (11 primiparous and 23 multiparous; 10 Days in Milk (DIM) < 70 and 24 DIM > 70), 13 cows were in group ARM (6 primiparous and 7 multiparous; 2 DIM < 70 and 11 DIM > 70), and 13 cows were in group SCM (2 primiparous and 11 multiparous; 3 DIM < 70 and 10 DIM > 70).

### 2.2. Extracellular Vesicles Isolation and RNA Extraction

For EVs isolation, the teats were cleaned with disposable wipes and disinfected (ethanol, 70%) before milking; then, the first flow of milk from each quarter was discarded, and a pooled sample of about 100 mL of milk for each cow was collected in sterile Falcon tubes, which were kept on ice during the collecting period and stored at −80 °C until analysis.

EVs were isolated using a modified isoelectrical precipitation method [45]. Briefly, frozen milk samples were thawed at room temperature and centrifuged at 2000× *g* at 4 °C for 10 min to separate fat. After elimination of the supernatant fat, the samples were centrifuged at 12,000× *g* at 4 °C for 40 min to remove cell debris. The supernatant was diluted 1:1 with distilled water and heated to 37 °C for 10 min in a water bath after adding HCl 6N to reach pH 4.6 for casein precipitation. Samples were centrifuged at 5000× *g* at room temperature for 20 min, and the supernatant was transferred into clean tubes and frozen at −80 °C overnight. After thawing, the samples were filtered with Millipore membrane filters of 1.00, 0.45, 0.20 μm (Merck Life Science, Milan, Italy), and the filtrate was centrifuged at 100,000× *g* at 4 °C for 1 h.

An aliquot of the pellet was resuspended in 0.1 M PBS pH 7.4 for the exosome characterization analysis. The Nanosight (Malvern Panalytical, Malvern, UK) under light scatter mode was used to visualize size distribution of the isolated milk exosomes in the 50–250 nm range. Transmission electron microscopy (TEM) with the immunogold labeling method was used to directly detect exosomes based on their size and specific surface proteins. Nickel TEM grids (Electron Microscopy Sciences, Hatfield, PA, USA), 400 mesh, with a formvar/carbon film, were floated on a drop of the pellet suspension fixed in 4% paraformaldehyde. Immunogold labeling was performed as previously described [26] using anti-HSC70 (ab19136, Abcam, Cambridge, MA, USA) and CD63 (ab193349, Abcam, Cambridge, MA, USA) primary antibodies coupled with a 10 nm gold particle (EY Lab. Inc., San Mateo, CA, USA). The grids were then washed with several drops of water and stained with methyl cellulose-uranyl acetate (4% uranyl acetate and 2% methyl cellulose in a ratio of 1:9) before being subjected to microscopic analysis. Grids were analyzed on Philips CM10 and images recorded at 80 kV. The results of TEM confirmed the presence of round-shaped exosomes with diameters of about 80 nm.

Another aliquot of the pellet was suspended in 300 μL of mirVana™ miRNA Isolation Kit lysis buffer (Ambion, ThermoFisher Scientific, Milan, Italy) and frozen at −80 °C. Extraction of miRNAs from the isolated EVs was performed with the commercial kit mirVana™ (ThermoFisher Scientific, Milan, Italy). The lysate was then mixed with 30 µL of homogenate additive and incubated on ice for 10 min and extracted once with 300 µL of acid phenol:chloroform. Samples were then centrifuged, and the aqueous (upper) phase was collected in a new tube. After washing in 100% absolute ethanol at room temperature, the lysate–ethanol mixture was placed in a filter cartridge and centrifuged until all the lysate solution had passed through the filter. The filter cartridge was then washed repeatedly until 40 µL of RNase-free water preheated to 95 °C was added. MiRNAs were collected by spinning for 25 s at 10,000× *g*.

### 2.3. miRNA Sequencing

Ten samples of the H group, 11 samples of the ARM group, and 11 samples of the SCM group were sequenced. RNA purity, integrity, and concentration were determined using an Agilent 2100 Bioanalyzer (Agilent Technologies, Santa Clara, CA, US). FASTQ raw sequence files were subsequently quality-checked with FASTQC software [46]. Then, sequences with a low quality score Q < 20 or including adaptor dimers were trimmed using Cutadapt 4.2 software [47]. Samples were selected on the basis of an RNA Integrity Number (RIN) of >7 and an rRNA 28 s/18 s ratio of >1.8. Libraries were prepared with Illumina^®^ TruSeq^®^ Small RNA Library Prep protocol (Illumina Inc., San Diego, CA, USA); 1 μg of total RNA for each library (minimum concentration of 200 ng/μL) was sequenced to 50 bp long single reads (average of 64 million reads per sample) using 8 lanes in 6-plex on an Illumina HiScanSQ (Illumina Inc., San Diego, CA, USA). Sequences were mapped against Btau_5.0.1 using Bioconductor Rsubread [48]. The raw data in FASTQ format were uploaded to NCBI Sequence Read Archive (Bioproject ID PRJNA902552).

### 2.4. Data Mining and Bioinformatic and Statistical Analysis

The resulting miRNA sequences were firstly divided into those annotated in *Bos taurus* (no. 184) and those in other vertebrate taxa (no. 622), whilst sequences annotated to other phyla were not included in the downstream analysis. To discover which miRNAs annotated in various vertebrate organisms overlapped in *Bos taurus*, sequences were searched in the miRbase [49]. The final number of miRNAs considered for *Bos taurus* was 225. These miRNAs were uploaded on the miRNet suite [50] and normalized with the DESeq2 method [51] for the comparisons of H vs. SCM, ARM vs. SCM, and H vs. ARM. The target genes of these mRNAs for *Bos taurus* were found based on miRTarBase version 8.0 [52] and miRanda database [53]. The list of differentially expressed (DE) target genes (*p* < 0.05 after Benjamini and Hochberg correction for multiple testing) resulting from each comparison were enriched to the Function Explorer on the Kyoto Encyclopedia of Genes and Genomes (https://www.kegg.jp/, accessed on 14 December 2022) database to find the statistically relevant pathways linked to the DE miRNAs and genes. Only enrichments with a *p*-value of <0.05 after Benjamini and Hochberg correction are reported.

## 3. Results

The characterization of the isolated milk exosomes was obtained by Nanoparticle Tracking Analysis (NTA), and the results are displayed as averaged finite track-length adjusted (FTLA) concentrations (Appendix A). Under electron microscopy, the exosomes were round with a diameter of approximately 80 nm (Appendix A). Moreover, the exosomes were positive for known exosomal markers CD63 and HSC70 via immunogold labeling.

The total number of reads from RNAseq obtained from 32 samples was 63,219,487, but after the quality check, 28,199,543 were annotated as RNAfam, and 18,507,870 were detected as miRNAs. A value of 656,318 counts per million (CPM) was calculated for miRNA followed by 196,430 CPM for RNA (Figure 1), and the contribution of other RNA types was minimal.

In the miRNet workflow, DESeq2 was applied to assess the DE of miRNAs in the different health states (H_ARM, ARM_SCM, and H_SCM). A total of 38 (16 upregulated and 22 downregulated), 18 (5 upregulated and 13 downregulated), and 12 (3 upregulated and 9 downregulated) miRNAs were significantly DE in the H_ARM, ARM_SCM, and H_SCM comparisons (Appendix A). The Venn diagram (Figure 2) showed that 1 DE miRNA was shared between the three groups (bta-mir-221), 1 DE miRNA in the H_SCM comparison (bta-mir-1247-5p), 9 DE miRNAs in the ARM_SCM comparison (bta-mir-142-5p; bta-mir-128; bta-mir-671; bta-mir-19a; bta-mir-146b; bta-mir-222; bta-mir-142-3p; bta-mir-15b; and bta-mir-205), and 21 DE miRNAs in the H_ARM comparison (bta-mir-2892; bta-mir-96; bta-mir-1343-3p; bta-mir-194; bta-mir-1281; bta-mir-2885; bta-mir-30c; bta-mir-1307; bta-mir-2415-3p; bta-mir-365-3p; bta-mir-542-5p; bta-mir-502°; bta-mir-7857-3p; bta-mir-30b-5p; bta-mir-2431-3p; bta-mir-328; bta-mir-210; bta-mir-186; bta-mir-31; bta-mir-1839; and bta-mir-425-3p).

The target genes of the DE miRNA were 2195 for the H_ARM comparison, 713 for the H_SCM comparison, and 1313 for the ARM_SCM comparison (Appendix A). Based on these genes, analysis of pathways revealed 57, 47, and 56 pathways significantly affected in the H_ARM, ARM_SCM, and H_SCM comparisons, respectively (*p* < 0.05 after HB correction for multiple testing) (Appendix A). A comparison of these enriched pathways from healthy, subclinical mastitic, and mastitis-prone cows showed that 19 pathways were differentially expressed in the three groups (Figure 3), while 16 were expressed only in the H_SCM group, 12 in the H_ARM comparison, and 18 in the ARM_SCM comparison.

Based on the degree and betweenness results, the top miRNAs were highlighted for each network (Appendix A). In the H_ARM network, bta-mir-2415-3p, bta-mir-2431-3p and bta-mir-6517, bta-mir-339a, bta-mir-1343-3p, bta-mir-2407, bta-mir-328, and bta-mir-1306 are reported, and these miRNAs were downregulated in the healthy cows compared with the at-risk cows. In the H_SCM pairwise comparison, the miRNAs bta-mir-339a, bta-mir-1247-5p and bta-mir-1306, bta-mir-345-5p, bta-mir-320a, and bta-mir-2388-3p were downregulated in the healthy cows compared to the cows with subclinical mastitis.

The significantly enriched KEGG signaling pathways are also shown in Figure 4, highlighting the NF-KB signaling pathway, T cell receptor signaling pathway, TNF signaling pathway, adherens junction, and leukocyte transendothelial migration for H_ARM. For H_SCM, the NF-KB signaling pathway was not enriched and was replaced by the NOD-like receptor signaling pathway. Figure 4A,B depict the connections between the miRNAs and target genes according to the selected KEGG pathways. They show the relevance of one miRNA in relation to the other miRNAs in regulating the relevant genes and pathways for immune functions.

## 4. Discussion

The analysis of miRNAs was performed in milk samples collected from a single milking and revealed 225 unique miRNAs associated with *Bos taurus* or showing sequence homology. The total number of miRNAs in the isolated milk exosomes was higher than that reported by Saenz-de-Juano et al. [31] and lower than that reported by Sun et al. [33], Cai et al. [30], and Ma et al. [54], and is likely dependent on library construction, reference *Bos taurus* genome and sequence depth, and isolation techniques.

In the present study, the mammary glands health status of the lactating cows was assessed by total (SCC) and differential counts (DSCC) of somatic cells in milk [18]. This method is routinely used by the Italian Breeders Association to classify lactating cows as healthy, affected by subclinical or chronic mastitis, or at risk of mastitis, based on validated cutoffs [22]. The leukocyte counts and miRNA cargos of exosomes in milk were measured in milk samples from healthy and mastitis-affected cows collected during a single milking. In this context, Saenz-de-Juano et al. [31] reported that the miRNA in exosomes and SCC did not vary on consecutive days and thus could reflect the condition of the individual cow at the level of the mammary gland.

The results in Figure 2 and in the Appendix A clearly show that the number of DE miRNAs from healthy cows (H) was higher compared with the ARM groups (21 miRNAs) than with the SCM groups (1 miRNA). Among the different studies on the cargos of isolated milk EVs, only four investigated the effects of mastitis on the presence and expression of miRNAs, three of them referred to *Staphylococcus aureus* infection, and only one compared subclinical with clinical mastitis. In particular, Ma et al. [54] collected milk samples from the four quarters of cows infected with this bacterial species, and Cai et al. [30] collected milk from the udder after infusion of *Staphylococcus aureus* colonies. Sun et al. [33] also infected the mammary gland with this pathogen, but only in two quarters, and used the corresponding quarters of individual cows as negative controls. In the fourth study [31], milk from cows with subclinical mastitis was collected from the infected quarters and compared with milk from the quarters with low SCC. The DE miRNAs in healthy and diseased cows were not the same in all four studies, except bta-miR-142-5p, which was always upregulated, not only in the cows infected with *Staphylococcus aureus* but also in the milk exosomes isolated from the milk of cows with subclinical mastitis. Our results showed that this miRNA was not DE compared with healthy cows (H_ARM, H_SCM), whereas it was downregulated in cows in the early phase of inflammation (ARM) compared with cows with subclinical mastitis (SCM) (*p* < 0.01). The upregulation of bta-miR-142-5p in cows with subclinical mastitis (SCC > 200,000/mL) compared with at-risk cows (SCC < 200,000/mL) is consistent with the results of Saenz-de-Juano et al. [31] and with the results of other studies in which cows were at the onset of mastitis after infection with *Staphylococcus aureus*. However, only one target gene for bta-miR-142-5p was found in cattle in the miRbase database. Interestingly, bta-miR-1247, the only upregulated miRNA in the H_SCM comparison, has been related to inflammation in bovine endometrial stromal cells treated with lipopolysaccharide affecting the MAPK pathway, which was significantly found in an H_ARM comparison [55]. Among the downregulated miRNAs in H_SCM and H_ARM comparisons, bta-mir-339a and bta-mir-320 were negatively associated with lactogenic differentiation in bovine mammary epithelial cell culture [56].

Bta-mir-22 and mir-345 microRNAs have been recently identified as differentially expressed in bovine macrophages in response to *M. bovis* expression [57]. Their differential expression is suggested to regulate host gene expression to enhance pathogen survival. Mir-345 is also a methylation-sensitive microRNA involved in cell proliferation. In addition to being downregulated in macrophages in response to *M. tuberculosis*, it is hypermethylated in T cells from TB-infected cattle [57].

A few other miRNAs found in the present study agreed with previous results, namely, bta-miR-142-3p, bta-miR-221, and bta-miR-103 with Cai et al. [30] and bta-miR-22-3p with Saenz-de-Juano et al. [31]. In addition, bta-miR-146a, bta-miR-146b, bta-miR-223, bta-miR-2285b, bta-miR-378-b, and bta-miR1246 were common between the studies, but the overlap did not extend to all of these studies. However, bta-miR-146a was found to alleviate intestinal colitis in a mouse colitis model by activating NF-κB [58], and it was found to be differentially expressed in bovine mammary glands affected by mastitis [59,60]. The experimental conditions in these investigations differed from those in the present study, suggesting that there is a great deal of individual variability in terms of infection-specific change, as previously suggested by Saenz-de-Juano et al. [31]. Indeed, the immunological response and milk concentrations of biomarkers also differed according to the type of mastitis [61]. Nevertheless, genetic variants [62] and factors other than infection influence the cargos of EVs in the milk of lactating cows. Stressors such as group relocation and ambient temperature have been reported to modulate miRNA levels in milk EVs from dairy cows [26,28]. Since miRNAs are also involved in mammary gland plasticity and the synthesis of milk components, genetic background is another factor to be considered [63].

Notably, bta-mir-320 and bta-mir-345 show 100% homology with human sequences, whereas bta-mir-146a is 93.8%. Since it is known that miRNAs can be absorbed in the human intestine, the study of miRNAs in milk exosomes may be a new area to investigate in relation to human health.

Significantly enriched pathways (Appendix A) underscored a different pattern between subclinical and at-risk cows compared with healthy cows. In the H_SCM comparison, at least nine pathways were directly or indirectly involved in the cellular immune response (leukocyte transendothelial migration; prolactin signaling pathway; Rap1 signaling pathway; signaling pathways regulating pluripotency of stem cells; NOD-like receptor signaling pathway; regulation of actin cytoskeleton; VEGF signaling pathway; thyroid hormone signaling pathway; and thyroid hormone synthesis). From a biochemical point of view, these data support the evidence for the increase in leukocyte count in subclinical mastitis. In the H_ARM group, however, the pattern was different. In the ARM samples, an increase in the percentage of PMN and lymphocytes (DSCC) was observed, but the total leukocyte count (SCC) was not yet above the cutoff generally considered as inflammation (200,000 cells/mL). Among the signaling pathways of this comparison, at least 11 related to immune cell function and modulation of inflammation (phosphatidylinositol signaling system; lysosome; Fc epsilon RI signaling pathway; Jak-STAT signaling pathway; chemokine signaling pathway; platelet activation; NF-kappa B signaling pathway; inflammatory mediator regulation of TRP channels; inositol phosphate metabolism; sphingolipid signaling pathway; and HIF-1 signaling pathway). Moreover, the target genes of bta-mir-339a were enriched for 10 significant pathways, including the MAPK signaling pathway [64], as for bta-mir-1247 in the present study.

The different patterns observed in the two comparisons can be explained by the different health status of the mammary gland. In the H_SCM comparison, we compared healthy with subclinical mastitis conditions, and in SCM there were clear signs of inflammation with an increase in leukocyte count above 200,000 cells/mL [18,22]. Therefore, one can expect a prevalence of signaling pathways involved in cell immune responses.

The H_ARM comparison includes samples in the initial stages of inflammation (ARM). In these animals, overt inflammation had not yet started, and SCC was <200,000 cells/mL, while PMN content was increased. ARM status can develop in two ways: spontaneous healing or subclinical mastitis. The regulation of the immune and inflammatory response is crucial for the final outcome. Therefore, we can assume that at this stage genes regulating the activity of cells, including epithelial cells that are also able to secrete inflammatory mediators and antibacterial substances [65,66], may play a role leading to one of the two possible outcomes in terms of the success of the defense response and the virulence of the invading pathogens.

These results are supported by assessing the degree (number of connections of nodes) and the betweenness (a measure of the number of shortest paths through a node; this filter retains genes connecting clusters) from the miRNet analysis, which allows us to highlight the most relevant miRNAs for the H_ARM and H_SCM networks (Figure 4; Appendix A for the full list). The interconnections of these miRNAs with selected significantly enriched KEGG pathways highlighted in Figure 4 (T cell receptor signaling pathway, TNF signaling pathway, leukocyte transendothelial migration, adherens junction, and NF-KB signaling pathway) were based only on the target sequences of the genes. The H_ARM comparison was the most interconnected and showed that a complex network exists between the miRNAs and the target genes within the selected signaling pathways. In contrast, in the H_SCM comparison, the DE miRNAs formed separate gene networks, indicating a more focused regulatory role of miRNAs within each pathway. However, these miRNAs do not correspond to those reported in the limited studies of exosomes in bovine mastitis [30,31,33,54]. Searches of the miRbase database and the literature revealed limited information on the regulatory activity of these miRNAs in the inflammatory process and immune response. The expression of bta-miR-2431-3p, bta-miR-2415-3p, bta-mir-2407, bta-mir-345-5p, and bta-mir-328 was reported in an in vitro study of viral infection of a bovine kidney cell line, but these miRNAs were not significantly affected by treatment [67]. In another study of DE miRNAs in bovine testicular and ovarian tissues [68], bta-miR-6517, bta-miR-1343-3p, and bta-mir-2388-3p were highlighted, but their biological significance was not elucidated. A sequencing and annotation study reported the expression of bta-mir-671 and bta-miR-1306 [69], but their regulatory roles were not investigated.

The limitations of the present study are that the extracellular vesicles were isolated from whole milk and were not related to a specific pathogen but were referred to an inflammatory process. Moreover, potential confounding factors, such as parity and stage of lactation, were not evaluated. Nevertheless, the origin of the extracellular vesicles is another aspect that deserves attention, and further studies are needed to determine the role of miRNAs in the inflammatory response of the mammary gland and their potential effects on human health to reach more comprehensive conclusions.

## 5. Conclusions

Extracellular vesicles and their miRNA cargos in milk are considered a promising approach to study the complex molecular machinery set in motion in response to mastitis in dairy cows and could influence milk quality. However, the question of whether miRNAs are influenced only by mastitis or also depend on various experimental conditions and environmental factors deserves further investigation.

## Figures and Tables

**Figure 1 animals-13-00821-f001:**
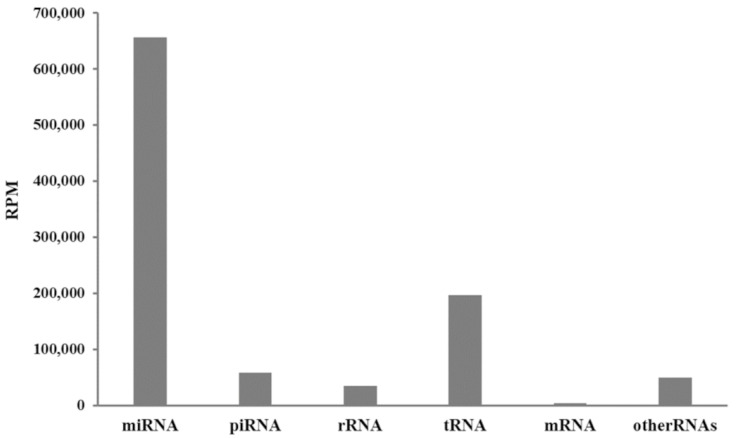
Mean small RNA and microRNA (miRNA) abundance in reads per million (RPM) of the milk samples collected from healthy and mastitic cows.

**Figure 2 animals-13-00821-f002:**
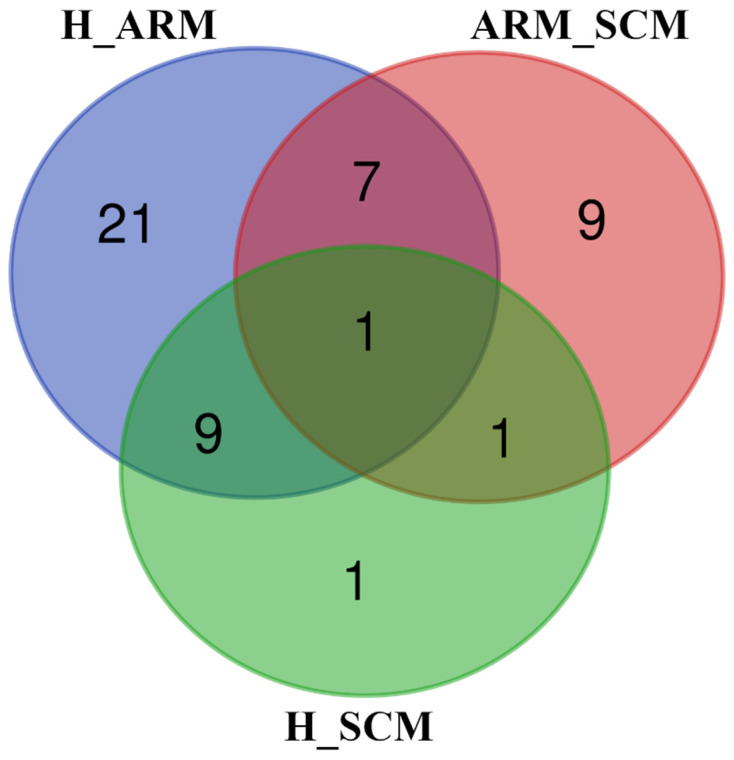
Venn diagram showing the number of unique or shared differentially expressed miRNAs in milk exosomes sampled from healthy (H) cows, cows with subclinical mastitis (SCM), and cows at risk of mastitis (ARM). H_ARM, H_SCM, and ARM_SCM refer to the pairwise comparisons between groups (Appendix A). Venn diagram was created by a web tool available at http://bioinformatics.psb.ugent.be/webtools/Venn/, accessed on 14 December 2022.

**Figure 3 animals-13-00821-f003:**
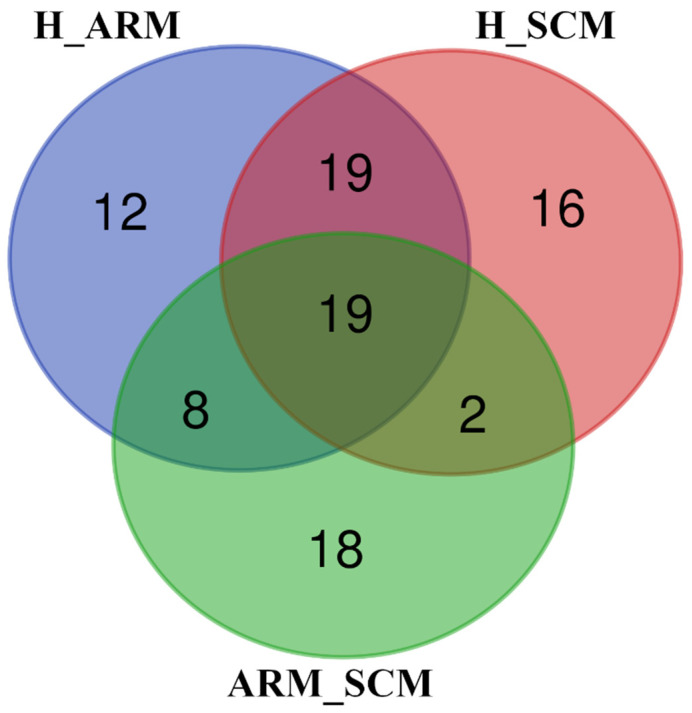
Venn diagram showing the number of unique or shared KEGG pathways of the target genes significantly enriched in the comparisons between healthy (H) cows, cows with subclinical mastitis (SCM), and cows at risk of mastitis (ARM). H_ARM, H_SCM, and ARM_SCM refer to the pairwise comparisons between groups (Appendix A). Venn diagram was created by a web tool available at http://bioinformatics.psb.ugent.be/webtools/Venn/, accessed on 14 December 2022.

**Figure 4 animals-13-00821-f004:**
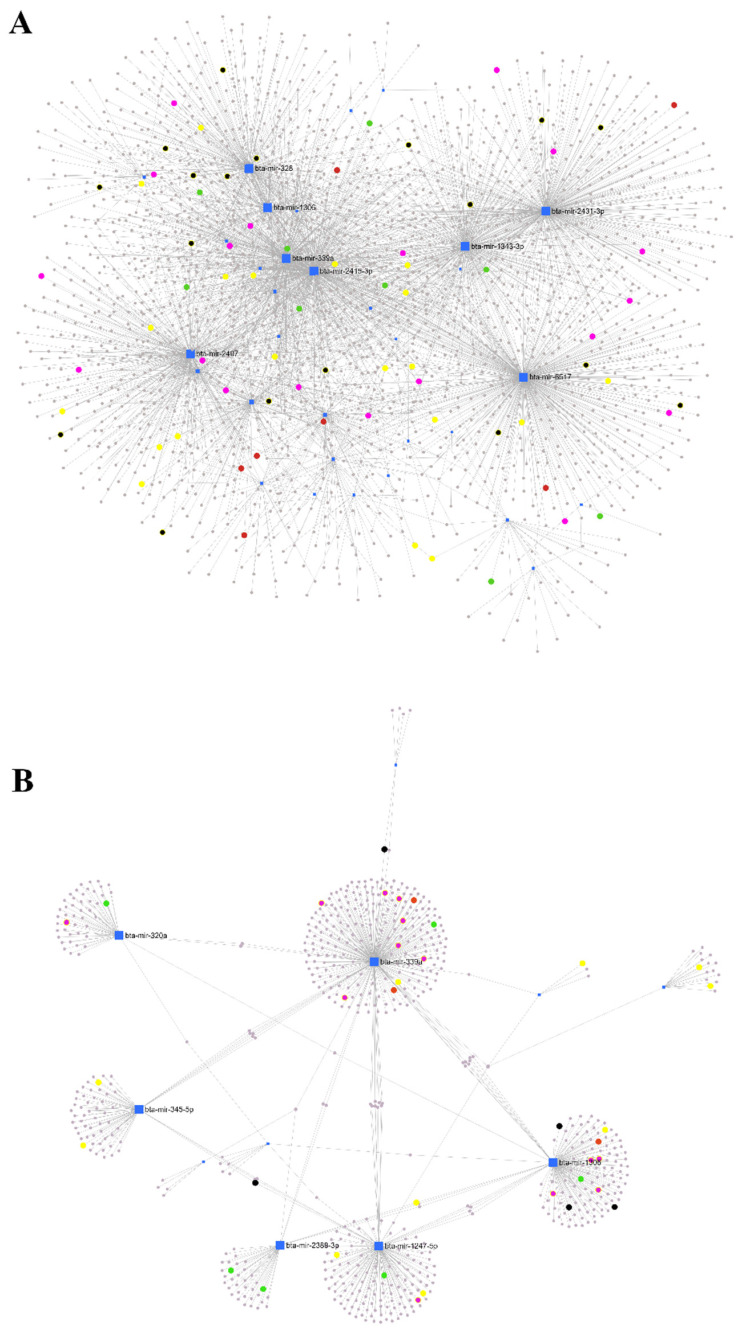
Network analysis of miRNA differentially expressed between healthy (H) cows, cows at risk of mastitis (ARM), and cows with subclinical mastitis (SCM). Figures were generated by miRNet suite [50]. (**A**) H_ARM comparison: in red are the genes of the NF-KB signaling pathway; in green the genes of the T cell receptor signaling pathway; in yellow the genes of the TNF signaling pathway; in purple the genes of the leukocyte transendothelial migration; and in black the genes of the adherens junction. (**B**) H_SCM comparison: in red are the genes of the NOD-like signaling pathway; in green the genes of the T cell receptor signaling pathway; in yellow the genes of the TNF signaling pathway; in purple the genes of the leukocyte transendothelial migration; in black the genes of the adherens junction. (Appendix A reports the miRNAs in the networks).

## Data Availability

Raw data FASTQ can be downloaded from NCBI Sequence Read Archive (https://www.ncbi.nlm.nih.gov/sra/, accessed on 1 December 2022), Bioproject ID PRJNA902552). Further inquiries can be directed to the corresponding author.

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
