# Peer review of "Regulatory Role of microRNA of Milk Exosomes in Mastitis of Dairy Cows"

_animals, 2023, doi:10.3390/ani13050821_

Round 1

Reviewer 1 Report

The study discusses an important topic, i.e., milk exosome miRNA and its role in relation to mammary infection in dairy cows. However, I have a major concern regarding this study.

The criteria for classifying the animals to different stages of mammary infection based on SCC, the percentage of neutrophils doesn’t seem appropriate to me.  The author classified their experimental animals as follows; healthy (group G, SCC/mL < 200,000 and DSCC ≤69.3%); at risk (group Y, SCC/mL < 200,000 and DSCC > 69.3%); with subclinical mastitis (group R, SCC/mL > 200,000 and DSCC > 69.3%). The authors also stated that DSCC is an estimate of the percentage of polymorphonuclear cells. Here I want to bring to the knowledge of the authors that the percentage of neutrophils in healthy milk is up to 30 % and reaches up to 45-50 % during subclinical mastitis and reaches 70 % only when there is clinical mastitis infection. Also, the number of SCC in milk up to 250,000 cells/mL doesn’t necessarily reflect mammary inflammation and could be easily attributed to many reasons including the productivity of the animal, stage of lactation, parity, season, time of the day, etc (This information all is missing in this manuscript). SCC values from 250,000 to 500,000 cells/mL are subclinical and beyond that is Clinical mastitis. In the current experimental design, it seems there was an improper selection of the experimental animals. Proper justification for grouping the animals into different groups based on the health status of their udder is needed.

Another important point: did you sample the animals for several days to make sure that the SCC is in this range before you took the final samples for analysis? This is because the SCC up to 250,000 cells/mL is not stable and you could find a completely different reading if sample the animal on the second day.

I understood from the methodology that the milk was not collected from the affected quarter or the quarter that was suspected to have a subclinical infection but from all the quarters and mostly the samples were healthy. Maybe the observed difference in the expression is attributed to other factors and not to mammary infection. Please clarify this more in the methodology and justify the same.

Also, the abbreviation for the name of the group is confusing. Please abbreviate the healthy group as (H) or simply use healthy. Also, for the subclinical mastitis group use SCM. Also, for the group at risk, use more relevant abbreviations so the reader can understand these abbreviations better and use them to read your manuscript more easily.

The author wrote “The Nanosight (Malvern Panalytical, UK) under light scatter mode was used to visualize size distribution of the isolated milk exosomes in the 50-250 nm  range. Transmission electron microscopy with the immunogold labelling method was used to directly detect exosomes based on their size and specific surface proteins”. This part should also be added to the results part under “characterization of the isolated milk exosomes”. You can add a figure showing the size distribution of exosomes under light scatter as well as photos of the Transmission electron microscopy.

The authors wrote, “The results of TEM confirmed the presence of round-shaped exosomes with diameters of about 80 nm and were presented in a previous review paper”. I can not see these results in the manuscript. As I said please add this information properly in the results section and document it with proper figures and photos etc.

Please provide more information about the extraction of miRNAs from the isolated EVs.

In the caption of the figures, name the software used to generate them just like in Figures 3, and 4. 

Reviewer 2 Report

Stefanon et al. reported some important miRNAs related to potential mastitis infection using the miRNAs sequencing from milk exosomes. Overall, the manuscript is informative and might be important to readers in the field. The manuscript has a limitation of sampling as miRNA levels vary among the lactation stage which has not been considered here.

Some minor suggestions are below.

Line 11: Define the abbreviation for miRNAs.

Line 13: affected or infected?

Line 14, 19, and others: The authors should define the G group, it is better to use three words for abbreviations such as NOR or normal, or healthy.

Line 13-17: The authors should focus on key miRNAs as markers (such as bta-mir-221) rather than the genes since the genes are only predicted and not validated.

Line 40: Define the abbreviation for WHO? And add a dot at the end of the sentence.

Line 65-66: The authors might add some sentences, and references to extend the introduction of key miRNAs in mastitis.

Line 113, 117, and others” be consistent in writing the company names.

Line 142: The authors might also deposit the raw count table for different groups in the supplementary files, so other researchers might benefit from meta-analyses.

Line 150: Which methods did the authors choose for preprocessing of raw counts and normalization of data?

Line 155-156: Some abbreviations are repeated such as DE.

Line 241-243; Line 309-313: Move it to the previous paragraph since it is not normal to get one sentence in the paragraph.

Line 342-347: The authors might move the discussion of limitations to the end of the discussion. Keep the conclusion concise.

Round 2

Reviewer 1 Report

Dear authors,

Thanks for your efforts to answer all questions and provide justifications for the inquiries.  After careful review, i regret to inform you that I cannot approve your research manuscript due to major defaults in the experimental design.

One of the main issues with the manuscript is the criteria used to classify the animals into different stages of mammary infection based on SCC and the percentage of neutrophils. The classification of healthy, at risk, and subclinical mastitis groups seems inappropriate to me. The percentage of neutrophils in healthy milk is usually up to 30% and increases to 45-50% during subclinical mastitis and 70% during clinical mastitis infection. Additionally, SCC values up to 250,000 cells/mL do not necessarily reflect mammary inflammation and could be due to various other

Another concern is the methodology used to sample the animals. The authors only sampled the animals once and did not observe SCC for several days to ensure the SCC was in a particular range before taking the final samples for analysis. This is important because SCC values up to 250,000 cells/mL are not stable and can vary greatly from day to day.

Lastly, the authors did not collect milk from the affected quarter, or the quarter suspected of having a subclinical infection, but from all quarters and mostly the samples were healthy. This might attribute the observed difference in expression to other factors rather than mammary infection.
